# Diffusion LLMs Can Do Faster-Than-AR Inference via Discrete Diffusion Forcing

**Xu Wang**[1][*], **Chenkai Xu**[1][*], **Yijie Jin**[1,3], **Jiachun Jin**[1], **Hao Zhang**[2], **Zhijie Deng**[1][†]

[1]Shanghai Jiao Tong University    [2]University of California San Diego    [3]Shanghai University

{wangxu60,132435xck,jiachun.jin,zhijied}@sjtu.edu.cn, jyj2431567@shu.edu.cn

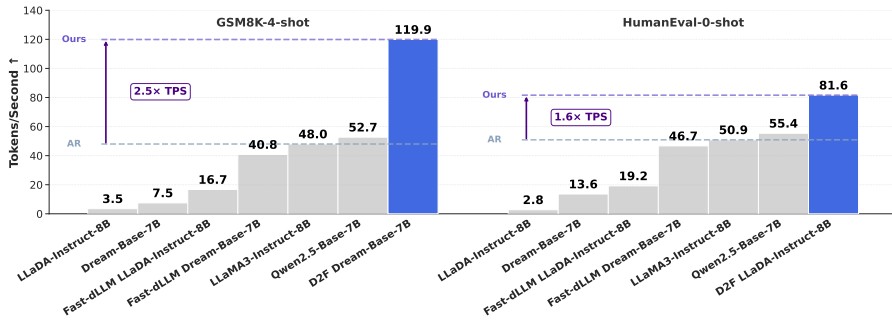

Figure 1: **D2F dLLMs surpass AR LLMs in inference speed for up to** $2.5\times$**.** Comparison of inference throughput among D2F dLLMs, vanilla dLLMs like Dream-Base-7B (Ye et al., 2025) and LLaDA-Instruct-8B (Nie et al., 2025), previous SOTA acceleration method Fast-dLLM (Wu et al., 2025), and similarly-sized AR baselines (Yang et al., 2024a; Grattafiori et al., 2024). The max generation length is set to 512.

## Abstract

Diffusion Large Language Models (dLLMs) have emerged as a promising alternative to autoregressive (AR) LLMs for text generation, with the potential to decode multiple tokens in a single iteration. However, none of the existing open-source dLLMs have achieved superior inference speed over AR LLMs of similar size. This paper breaks this barrier based on a simple and effective strategy named discrete diffusion forcing (D2F). D2F equips dLLMs with two key capabilities: (1) block-wise autoregressive generation to enable KV cache utilization; (2) prediction of following tokens without requiring completion of prior blocks for inter-block parallel decoding. In this way, the vanilla dLLMs are refurbished into an AR-diffusion hybrid paradigm for efficient inference. D2F can be implemented with an asymmetric distillation process based on pre-trained dLLMs to achieve rapid convergence. We further propose a pipelined parallel decoding algorithm, which enables a trade-off between efficiency and efficacy. Empirically, D2F dLLMs achieve more than $2.5\times$ inference speed than LLaMA3 and Qwen2.5 on GSM8K. Compared to vanilla dLLMs such as LLaDA and Dream, D2F delivers over $50\times$ acceleration while preserving comparable output quality. Moreover, it boosts DiffuCoder by up to $11.8\times$ speedup without sacrificing performance. Code is available at https://github.com/SJTU-DENG-Lab/Discrete-Diffusion-Forcing.

## 1 Introduction

Large Language Models (LLMs) have maintained a dominant position in text generation for a long time (Achiam et al., 2023; Touvron et al., 2023a; Yang et al., 2025; Grattafiori et al., 2024). Recently,

---

[*]Equal contribution.
[†]Corresponding author.

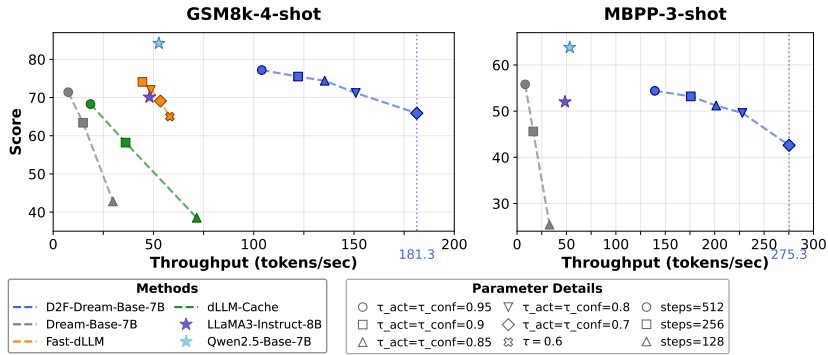

Figure 2: **Throughput vs. performance trade-off.** As shown, D2F achieves a more favorable trade-off compared to vanilla dLLMs. Refer to Section 4.3 for the details of the hyperparameters $\tau_{\text{add}}$ and $\tau_{\text{conf}}$.

Diffusion Large Language Models (dLLMs) have emerged as a promising alternative to LLMs (Ye et al., 2025; Nie et al., 2025; Zhu et al., 2025), acknowledged by their potential to generate multiple text tokens in parallel. For example, closed-source dLLMs such as Gemini Diffusion (Google DeepMind, 2025), Mercury (Inception et al., 2025), and Seed Diffusion (Song et al., 2025) can yield thousands of tokens per second, 5-10 times faster than autoregressive (AR) LLMs of similar size.

However, the speed merits of dLLMs have not been demonstrated within the open-source community. Approaches to bridge the gap include designing KV cache strategies (Arriola et al., 2025; Liu et al., 2025; Ma et al., 2025) and improving parallel sampling algorithms (Wu et al., 2025; Wei et al., 2025; Hu et al., 2025). For instance, Block Diffusion (Arriola et al., 2025) turns dLLMs into a block-wise sequential generation paradigm to leverage KV cache. Yet, it precludes the inter-block parallelism, a crucial factor for efficient inference. Fast-dLLM (Wu et al., 2025) also adopts a block-wise generation order to facilitate the reuse of states of generated tokens and incorporates confidence-based remasking for parallel decoding. Nonetheless, the cached states can be biased after subsequent tokens are decoded due to the bidirectional nature of the involved attention.

This paper achieves the first breakthrough in accelerating dLLMs to a faster-than-AR regime. Conceptually, we aim to embrace block-wise sequential generation to facilitate KV cache utilization, yet reject the dilemma that the decoding of subsequent blocks must wait for preceding blocks to be fully denoised. This implies a novel AR-diffusion hybrid paradigm, which, however, cannot be realized through naive teacher-forcing training. This is because teacher-forcing requires complete preceding information to predict subsequent content. Interestingly, this insight shares the spirit with the diffusion forcing (DF) (Chen et al., 2024a) technique developed specifically for continuous-space diffusion models. In this sense, this paper forms an extension of DF to discrete data, giving rise to the discrete diffusion forcing (D2F) method for dLLM acceleration.

Concretely, D2F dLLMs learn to denoise a sequence of token blocks with monotonically increasing mask ratios in parallel. Naturally, preceding blocks can finish before subsequent ones, allowing their KV states to be cached for subsequent computations. Note that we constrain the attention to be block-wise causal to ensure the KV cache remains accurate. For training efficiency, we distill D2F dLLMs from existing bidirectional attention dLLMs using an asymmetric distillation loss. In inference, we design a pipelined parallel decoding algorithm which enables inter-block parallelism and offers a decent trade-off between inference efficiency and performance (see Figure 2).

Distilled on the Bespoke-Stratos-17k (Bespoke Labs, 2025) for 12 hours with 8 NVIDIA A100-SXM4-40GB GPUs, D2F can accelerate LLaDA-Instruct-8B (Nie et al., 2025) and Dream-Base-7B (Ye et al., 2025) for over $50\times$ without degradation on mathematical and programming benchmarks, including GSM8K (Cobbe et al., 2021), MATH (Hendrycks et al., 2021), HumanEval (Chen et al., 2021) and MBPP (Austin et al., 2021b). More importantly, D2F-Dream-Base-7B is up to $2.5\times$ faster than LLaMA3-Instruct-8B (Grattafiori et al., 2024) on GSM8K and $1.6\times$ faster on HumanEval, establishing the first open-source dLLMs that outrun AR ones. Moreover, D2F can accelerate DiffuCoder (Gong et al., 2025) by up to $\mathbf{11.8\times}$ without sacrificing performance.

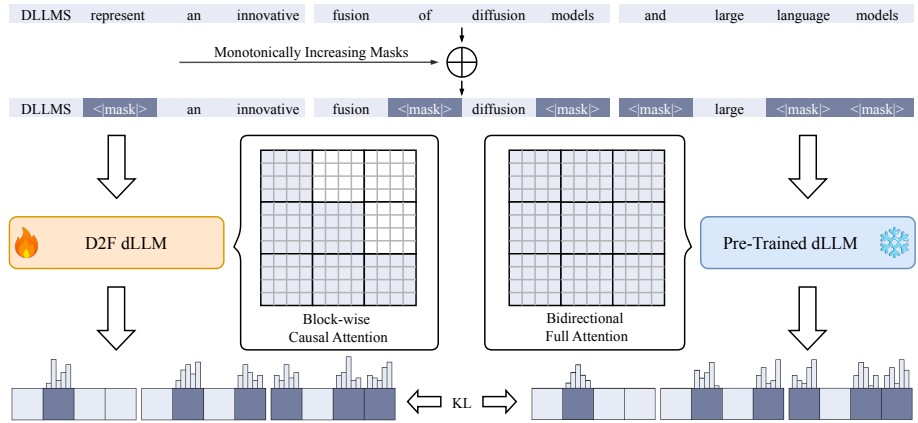

Figure 3: **An overview of discrete diffusion forcing (D2F).** During training, the answer is divided into blocks with progressively increasing masking ratios. D2F dLLM is specified with a block-wise causal attention mask, and trained to mimic the prediction of a pre-trained bidirectional dLLM conditioned on partially denoised preceding tokens. This enables inter-block parallel decoding with KV cache compatibility during inference.

In summary, this work makes the following key contributions:

- **Discrete Diffusion Forcing (D2F),** a novel framework adapting diffusion forcing to the discrete domain. Unlike prior work, D2F is specifically designed to unlock inter-block parallelism for inference acceleration.
- **An asymmetric distillation strategy** that efficiently refurbishes pre-trained bidirectional dLLMs into block-wise causal models, avoiding the high cost of training from scratch.
- **A pipelined parallel decoding algorithm** which leverages accurate KV caching to decode future blocks before prior ones are fully generated, ensuring high throughput.
- **The first faster-than-AR dLLMs.** Our models achieve up to 2.5× speedup over LLaMA3 (Grattafiori et al., 2024) and Qwen2.5 (Yang et al., 2024a) on GSM8K, establishing a new speed benchmark for open-source dLLMs.

## 2 RELATED WORK

**Diffusion Large Language Models (dLLMs).** The landscape of language generation has long been defined by autoregressive models (Achiam et al., 2023; Guo et al., 2025; Liu et al., 2024). These models, renowned for their high-quality output, are inherently limited by a sequential, token-by-token decoding process. To overcome this latency bottleneck, dLLMs have emerged (Ye et al., 2025; Nie et al., 2025; Google DeepMind, 2025; Inception et al., 2025; Zhu et al., 2025; Gong et al., 2025). Instead of generating text sequentially, dLLMs operate by iteratively denoising a fully masked sequence, a process that enables the parallel prediction of all tokens at once. This approach, which draws from non-autoregressive principles, utilizes a bidirectional attention mechanism to achieve a more holistic understanding of context. Recent large-scale dLLMs, such as LLaDA (Nie et al., 2025), trained from scratch, and Dream (Ye et al., 2025), initialized from pre-trained AR weights, have demonstrated outstanding performance competitive with leading ARMs, establishing dLLMs as a powerful alternative paradigm for high-quality, parallelizable text generation.

**Acceleration of dLLMs.** dLLMs suffer from slower inference than autoregressive models due to incompatibility with standard KV cache and limited parallelization. Existing acceleration methods fall into two categories. First, caching-based approaches (Liu et al., 2025; Wu et al., 2025; Ma et al., 2025) develop approximate schemes to reuse computations for static sequence parts, as standard KV cache is incompatible with bidirectional attention. Second, sampling optimization methods reduce decoding steps through confidence-aware strategies (Wu et al., 2025), auxiliary model guidance (Israel et al., 2025), or adaptive decoding speeds (Wei et al., 2025). However, these methods achieve limited speedups due to inherent approximations and auxiliary model overhead, often failing to match the efficiency of comparable AR (Achiam et al., 2023; Touvron et al., 2023a; Yang

et al., 2025; Touvron et al., 2023b; Grattafiori et al., 2024) models. Our approach fundamentally restructures generation into a block-autoregressive framework that is compatible with standard KV cache, while enabling prediction of subsequent tokens without completing previous blocks.

**AR-diffusion hybrid models.** Recent studies have explored incorporating the speed advantages of autoregressive models into diffusion-based frameworks, particularly in tasks such as video generation (Yin et al., 2025; Po et al., 2025; Sun et al., 2025; Huang et al., 2025). For the video generation task, it is common to model the temporal dependencies between frames using an autoregressive approach, while applying denoising within each frame independently. This hybrid architecture leverages the acceleration benefits of KV cache enabled by the autoregressive modeling paradigm, while preserving the high generation quality of diffusion-based methods.

## 3 PRELIMINARY: DIFFUSION LARGE LANGUAGE MODELS (DLLMS)

Diffusion models, originally developed for continuous data, have achieved state-of-the-art results in image synthesis (Esser et al., 2024; Chen et al., 2024b; Podell et al., 2023; Labs et al., 2025) and video generation (Peng et al., 2025; Zheng et al., 2024; Yang et al., 2024b; Bao et al., 2024). In recent years, advances in the theory of discrete diffusion (Austin et al., 2021a; Shi et al., 2024; Lou et al., 2023; Campbell et al., 2022) have facilitated the emergence of large-scale dLLMs (Nie et al., 2025; Ye et al., 2025) for text generation tasks, which have demonstrated performance competitive with their AR counterparts, while offering the potential for parallel generation.

The majority of successful dLLMs operate under a masked diffusion paradigm (Nie et al., 2025). This process begins with a forward process, where an original text sequence of $L$ tokens, $Y^0 = \{y_1^0, \ldots, y_L^0\}$, is progressively corrupted into a noisy sequence $Y^t$ over a continuous time schedule $t \in [0, 1]$. This corruption is routinely achieved by replacing original tokens independently with a special [MASK] token. Typically, we can define the conditional distribution as:

$$q(Y^t|Y^0) = \prod_{i=1}^{L} q(y_i^t|y_i^0), \quad \text{where} \quad q(y_i^t|y_i^0) = \begin{cases} 1 - t, & \text{if } y_i^t = y_i^0 \\ t, & \text{if } y_i^t = \texttt{[MASK]} \end{cases}. \tag{1}$$

Consequently, $Y^1$ represents a fully masked sequence. dLLMs are designed to learn a parameterized model $p_\theta(Y^0|Y^t)$ to reverse the forward process, hence enabling denoising from the fully masked sequence $Y^1$ to language samples $Y^0$. This formulation allows the model $p_\theta$ to predict all mask tokens in the sequence simultaneously at each inference step, forming the basis of dLLMs' theoretical potential to surpass the AR paradigm for the high-speed, parallel generation.

However, practical implementations of dLLMs face severe inference efficiency bottlenecks. First, the use of bidirectional attention mechanisms fundamentally conflicts with KV cache, leading to significant redundant computation across denoising steps. Second, the reliance on a conditional independence assumption for parallel decoding makes it hard to generate interdependent tokens (Song & Zhou, 2025), so more iterative steps are required for high-quality outputs. While prior works attempt to mitigate these issues, for instance, by introducing block-wise sequential generation to enable KV cache (Arriola et al., 2025) or implementing approximate KV cache (Wu et al., 2025; Liu et al., 2025), they fail to simultaneously achieve both precise KV cache and efficient parallel decoding. Consequently, no open-source dLLMs has yet matched the inference speed of AR models.

## 4 METHOD

The paper derives discrete diffusion forcing (D2F) to accelerate the inference of dLLMs to a faster-than-AR regime. Refer to Figures 3 and 4 for an overview of the training and inference of D2F.

### 4.1 DISCRETE DIFFUSION FORCING

D2F endows dLLMs with two critical capabilities: (1) *block-level AR generation*, which enables efficient standard KV cache and significantly reduces computational redundancy; and (2) *inter-block parallel decoding*, where the model is trained to predict future blocks from incomplete, partially reconstructed predecessors to maximize the number of decoded tokens per inference step. This naturally gives rise to an AR-diffusion hybrid modeling paradigm, but naive teacher-forcing training,

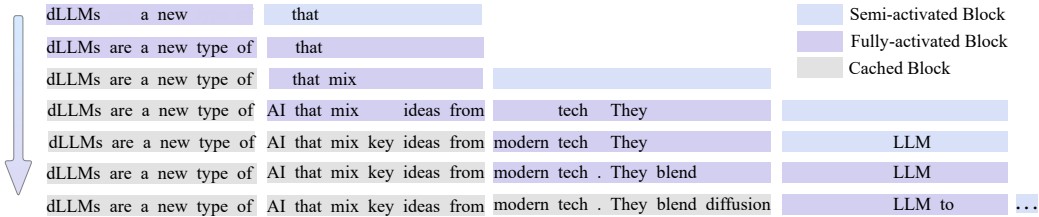

Figure 4: **Visualization of the pipelined parallel decoding of D2F dLLMs.** As shown, a pipeline of blocks are decoded in parallel. A new block is dynamically added when the completion ratio of the last block exceeds a threshold $\tau_{add}$ ($= \frac{1}{3}$ here). The new block is initially semi-activated and will transition to a fully-activated state when its predecessor reaches the completion threshold $\tau_{act}$ ($= \frac{5}{6}$ here). The fully-activated blocks differ from semi-activated ones in that they would decode multiple tokens in each step more aggressively.

---

**Algorithm 1** Asymmetric Distillation for D2F

---

**Require:** Pre-trained dLLM $p_{\phi^-}$; D2F dLLM $p_\theta$; block size $k$; training dataset $\mathcal{D}$.
1: **while** training **do**
2:     Sample a sequence $Y$ from $\mathcal{D}$.
3:     Divide $Y$ into $N$ blocks $\{Y_{B_1}, \ldots, Y_{B_N}\}$, each of size $k$.
4:     Sample a monotonic noise schedule $\{t_1, \ldots, t_N\}$ where $t_1 < \cdots < t_N$.
5:     For each $i \in \{1, \cdots, N\}$, corrupt $Y_{B_i}$ to $Y_{B_i}^{t_i}$ using Eq. 1.
6:     Predict distributions for each block $Y_{B_i}^0$ with:
7:         Teacher (global view): $p_{\phi^-}(Y_{B_i}^0 | Y_{B_1}^{t_1}, \ldots, Y_{B_N}^{t_N})$
8:         Student (causal view): $p_\theta(Y_{B_i}^0 | Y_{B_1}^{t_1}, \ldots, Y_{B_i}^{t_i})$
9:     Update $p_\theta$ based on the loss $\mathcal{L}_{\text{D2F}}$ defined in Eq. 3.

---

as employed in block diffusion (Arriola et al., 2025), cannot lead to the second capability. To address this, D2F trains the model to perform conditional denoising of the current block based on a partially denoised prefix, enabling more coherent and efficient sequential generation.

Concretely, D2F partitions a clean sequence $Y^0$ into $N$ blocks of size $k$. Let $B_i := \{(i-1)*k, \ldots, i* k - 1\}$ denote the token indices in the $i$-th block and $Y_{B_i}$ denote the corresponding subsequence. In the forward process, D2F applies a monotonically increasing noise schedule ($t_1 < t_2 < \cdots < t_N$) to the $N$ blocks, i.e., $Y^t = \{Y_{B_1}^{t_1}, \ldots, Y_{B_N}^{t_N}\}$. Namely, the earlier blocks in $Y^t$ are progressively less masked (i.e., more complete), while later blocks remain increasingly masked (i.e., more uncertain). For the reverse process, D2F trains a $\theta$-parameterized model to characterize:

$$p_\theta(Y^0|Y^t) = \prod_{i=1}^{N} p_\theta(Y_{B_i}^0 | Y_{B_1}^{t_1}, \ldots, Y_{B_i}^{t_i}). \tag{2}$$

Intuitively, the learned model can first complete the decoding of preceding blocks while simultaneously advancing the denoising of subsequent ones, which effectively enables inter-block parallel decoding. By preserving a causal attention structure across blocks—wherein intra-block attention remains bidirectional—we can cache the KV states of already decoded blocks for exact reuse, thereby reducing redundant computations and improving inference efficiency.

**Connection to diffusion forcing (Chen et al., 2024a).** From a high-level perspective, our approach bears a strong conceptual resemblance to diffusion forcing (DF) (Chen et al., 2024a), originally developed for continuous-space diffusion models and notably applied in video generation (Yin et al., 2025). Both methods involve predicting the tokens of the next block conditioned on a noisy, incomplete premise, and our framework can be viewed as an extension of DF to discrete sequences. Such a principled extension motivates our naming of *discrete diffusion forcing*.

### 4.2 ASYMMETRIC DISTILLATION

Noting that training a dLLM with billions of parameters from scratch can be costly (Nie et al., 2025; Ye et al., 2025), we propose to distill a D2F dLLM from a pre-trained vanilla dLLM available in

---

**Algorithm 2** Pipelined Parallel Decoding for D2F Inference

---

**Require:** D2F model $p_\theta$; thresholds $\tau_{\text{add}}, \tau_{\text{act}}, \tau_{\text{conf}}$.
1: Initialize $Y = \{Y_{B_1}\}$ with a block of mask tokens.
2: **while** generation is not complete **do**
3:     **if** the ratio of decoded tokens in $Y_{B_{-1}}$ exceeds $\tau_{\text{add}}$ and $<|\text{EOS}|>$ not in $Y$ **then**
4:         Append a new fully masked block with semi-activated state to $Y$.
5:     **for** the active block $Y_{B_i}$ in $Y$ **do**
6:         **if** the ratio of decoded tokens in $Y_{B_{i-1}}$ exceeds $\tau_{\text{act}}$ **then**
7:             Set $Y_{B_i}$ to be fully-activated
8:     Forward pass of $Y$ with D2F dLLM $p_\theta$ using cached KV
9:     **for** the active block $Y_{B_i}$ in $Y$ **do**
10:        Let $J_i$ record the set of token positions in $B_i$ with $> \tau_{\text{conf}}$ prediction confidence
11:        **if** $Y_{B_i}$ is fully-activated and $J_i$ is $\emptyset$ **then**
12:            Add the token position with the highest confidence to $J_i$
13:        Sample tokens with positions in $J_i$ and remask other positions
14:     Update KV cache for completed blocks

---

the open-source community. Let $p_{\phi^-}$ denote the standard bidirectional teacher dLLM and $p_\theta$ the student D2F dLLM ($\theta$ is initialized as $\phi^-$). The distillation minimizes the following loss

$$\mathcal{L}_{\text{D2F}} = \mathbb{E}_{t_1 < \cdots < t_N} \left[ \sum_{i=1}^{N} D_{\text{KL}} \left( p_\theta(Y_{B_i}^0 | Y_{B_1}^{t_1}, \ldots, Y_{B_i}^{t_i}) \| p_{\phi^-}(Y_{B_i}^0 | Y_{B_1}^{t_1}, \ldots, Y_{B_N}^{t_N}) \right) \right], \quad (3)$$

where $D_{\text{KL}}$ represents the KL divergence aggregated over the mask tokens. As shown, the distillation is asymmetric—the teacher $p_{\phi^-}$ predicts for each block $Y_{B_i}^0$ with a global view of all noisy blocks while the student $p_\theta$ learns to approximate using only a causally restricted view. In this way, the mask prediction capabilities of existing dLLMs can be embedded into a new D2F dLLM. This objective also connects to CausVid (Yin et al., 2025), which distills existing holistic diffusion video generators into streaming ones. As for model architecture, the student differs from the teacher solely in attention masks—it uses the block-wise causal attention instead of a bidirectional one.

We summarize the whole algorithmic process in Algorithm 1.

### 4.3 Pipelined Parallel Decoding

As illustrated in Figure 4, we introduce a pipelined parallel decoding algorithm for the inference of D2F dLLMs. Specifically, we maintain a sliding window of active blocks and dynamically append a new fully-masked block when the decoding progress of the last block exceeds a threshold $\tau_{\text{add}}$. The dynamic strategy significantly reduces the per-step computational cost compared to maintaining a full sequence of massive blocks throughout inference.

Observing that aggressive decoding of a newly added block can degrade performance, we incorporate a dual-state decoding mechanism. Concretely, the newly added block is initialized in a semi-activated state to enable conservative parallel decoding, and will become fully activated when its predecessor has finished $\tau_{\text{act}}$ of the decoding—i.e., sufficient contextual information has been accumulated to support aggressive decoding of the latter block. Following Fast-dLLM (Wu et al., 2025), semi-activated blocks admit tokens with confidence above a threshold $\tau_{\text{conf}}$, while fully activated blocks additionally enforce the selection of the most confident token when no such token exists.

The synergy between the dynamic block management and dual-state mechanism conjoins per-step efficiency and inter-block parallelism. It is interesting to note that our approach also shares conceptual similarities with prior work in video generation (Teng et al., 2025). More algorithmic details are provided in Algorithm 2 and a hyperparameter analysis is in Table 3.

## 5 Experiments

This section details the experimental setup and presents the results of D2F dLLMs.

| Test Set | Method | TPS ↑ | Latency (s) ↓ | Gen. Length | Score ↑ |
|---|---|---|---|---|---|
| **GSM8K**
4-shot | LLaDA-Instruct | 7.2 (1.0x) | 32.3 (1.0x) | 231 | 77.4 |
| | dLLM-Cache | 20.1 (2.8x) | 11.5 (2.8x) | 231 | 77.5 |
| | Fast-dLLM (Prefix-Cache) | 33.3 (4.6x) | 7.0 (4.6x) | 232 | 77.8 |
| | Fast-dLLM (Dual-Cache) | 35.2 (4.9x) | 6.6 (4.9x) | 232 | **78.9** |
| | **D2F-LLaDA** | **52.5 (7.3x)** | **2.8 (11.5x)** | 144 | 77.3 |
| **MBPP**
3-shot | LLaDA-Instruct | 0.9 (1.0x) | 71.4 (1.0x) | 65 | **39.0** |
| | dLLM-Cache | 2.3 (2.6x) | 28.3 (2.5x) | 66 | 37.0 |
| | Fast-dLLM (Prefix-Cache) | 13.0 (14.4x) | 4.9 (14.6x) | 64 | 37.6 |
| | Fast-dLLM (Dual-Cache) | 15.3 (17.0x) | 3.8 (18.8x) | 58 | 36.4 |
| | **D2F-LLaDA** | **47.6 (52.9x)** | **1.4 (51.0x)** | 68 | 38.0 |
| **HumanEval**
0-shot | LLaDA-Instruct | 2.8 (1.0x) | 38.8 (1.0x) | 107 | 36.0 |
| | dLLM-Cache | 4.5 (1.6x) | 23.3 (1.7x) | 104 | 39.0 |
| | Fast-dLLM (Prefix-Cache) | 13.7 (4.9x) | 7.4 (5.2x) | 102 | 38.4 |
| | Fast-dLLM (Dual-Cache) | 19.2 (6.9x) | 5.2 (7.5x) | 100 | 35.4 |
| | **D2F-LLaDA** | **81.6 (29.1x)** | **1.6 (24.3x)** | 133 | **40.2** |
| **Math**
4-shot | LLaDA-Instruct | 21.1 (1.0x) | 11.5 (1.0x) | 243 | 23.7 |
| | dLLM-Cache | 26.9 (1.3x) | 9.1 (1.3x) | 246 | 23.2 |
| | Fast-dLLM (Prefix-Cache) | 47.7 (2.3x) | 5.2 (2.2x) | 246 | 22.4 |
| | Fast-dLLM (Dual-Cache) | 42.5 (2.0x) | 5.8 (2.0x) | 246 | 22.4 |
| | **D2F-LLaDA** | **90.2 (4.3x)** | **4.3 (2.7x)** | 384 | **29.1** |

Table 1: **Performance comparison of various acceleration methods on LLaDA-Instruct-8B**. Speedup ratios are shown in (green). All baseline methods use the default sampling configuration from the original LLaDA implementation. See Appendix D for detailed hyperparameters.

## 5.1 EXPERIMENT SETTINGS

We perform evaluation on two representative dLLMs: LLaDA-Instruct-8B (Nie et al., 2025) and Dream-Base-7B (Ye et al., 2025)—backbones shared by prior acceleration work—for fair comparison. For our distillation-based training, we utilize a dataset derived from the Bespoke-Stratos-17k benchmark (Bespoke Labs, 2025). Specifically, we use two publicly available collections sourced from the HuggingFace Hub , where a third party had previously generated responses to the problems from Bespoke-Stratos-17k using the Qwen2.5-7B model (Yang et al., 2024a). These collections are pre-filtered to a maximum length of 600 tokens. The q_proj, v_proj, k_proj, and o_proj modules of D2F dLLMs are tuned with the LoRA (Hu et al., 2022) technique, using the rank of 32, the scaling factor of 32, and the dropout rate of 0.1. During training, We truncate or pad all sequences to a final length of 1024 tokens, and employ a block size of 16. We applied block-wise monotonically increasing mask ratios, with a maximum threshold of 0.7 and a minimum threshold of 0.2, inspired by Block Diffusion (Arriola et al., 2025). During inference, unless otherwise specified, $\tau_{conf}$ is set to 0.9, $\tau_{add}$ is set to 0.1, and $\tau_{act}$ is set to 0.95. The models are trained for 12 hours using an AdamW optimizer with a constant learning rate of $10^{-5}$. All training and inference are conducted on a setup comprising 8 NVIDIA A100-SXM4-40GB GPUs. Detailed hyperparameter configurations for inference on each benchmark are provided in the Appendix D. In addition, we conduct experiments on DiffuCoder (Gong et al., 2025), and report the results in Appendix E.

## 5.2 MAIN RESULTS

**Benchmarks.** Following established conventions, performance evaluation of D2F is conducted across mathematical reasoning and code generation benchmarks, including GSM8K (Cobbe et al., 2021), GSM8K-CoT (Chain-of-Thought reasoning variant of GSM8K), Math (Hendrycks et al., 2021), HumanEval (Chen et al., 2021), and MBPP (Austin et al., 2021b).

**Baselines.** Comprehensive comparisons are established between D2F and state-of-the-art acceleration strategies, including Fast-dLLM (Wu et al., 2025) and dLLM-Cache (Liu et al., 2025), implemented on LLaDA-Instruct-8B (Nie et al., 2025) and Dream-Base-7B (Ye et al., 2025) architectures.

| Test Set | Method | TPS ↑ | Latency (s) ↓ | Gen. Length | Score ↑ |
|---|---|---|---|---|---|
| **GSM8K-CoT**
8-shot | Dream-Base | 9.5 (1.0x) | 26.8 (1.0x) | 255 | 75.0 |
| | dLLM-Cache | 26.0 (2.7x) | 9.8 (2.7x) | 255 | 72.0 |
| | Fast-dLLM (Prefix-Cache) | 50.3 (5.3x) | 5.1 (5.3x) | 255 | 76.6 |
| | Fast-dLLM (Dual-Cache) | 49.8 (5.2x) | 5.1 (5.3x) | 255 | 75.0 |
| | **D2F-Dream** | **91.2** (9.6x) | **2.8** (9.6x) | 256 | **77.6** |
| **MBPP**
3-shot | Dream-Base | 10.4 (1.0x) | 24.6 (1.0x) | 256 | 56.2 |
| | dLLM-Cache | 25.5 (2.5x) | 10.0 (2.5x) | 256 | 52.6 |
| | Fast-dLLM (Prefix-Cache) | 71.6 (6.9x) | 3.6 (6.8x) | 256 | **56.4** |
| | Fast-dLLM (Dual-Cache) | 73.2 (7.0x) | 3.5 (7.0x) | 256 | 51.0 |
| | **D2F-Dream** | **105** (10.1x) | **2.3** (10.7x) | 240 | 55.2 |
| **HumanEval**
0-shot | Dream-Base | 20.2 (1.0x) | 12.6 (1.0x) | 255 | 54.3 |
| | dLLM-Cache | 23.2 (1.1x) | 11.0 (1.1x) | 255 | **55.5** |
| | Fast-dLLM (Prefix-Cache) | 62.4 (3.1x) | 4.1 (3.1x) | 255 | 54.3 |
| | Fast-dLLM (Dual-Cache) | 60.0 (3.0x) | 4.3 (2.9x) | 255 | 53.0 |
| | **D2F-Dream** | **73.2** (3.6x) | **3.1** (4.1x) | 227 | 54.3 |
| **Math**
4-shot | Dream-Base | 9.9 (1.0x) | 25.8 (1.0x) | 256 | 35.8 |
| | dLLM-Cache | 12.7 (1.3x) | 20.2 (1.3x) | 256 | 34.5 |
| | Fast-dLLM (Prefix-Cache) | 65.6 (6.6x) | 3.9 (6.6x) | 256 | **37.6** |
| | Fast-dLLM (Dual-Cache) | 67.0 (6.8x) | 3.8 (6.8x) | 256 | 37.1 |
| | **D2F-Dream** | **98.8** (10.0x) | **2.6** (9.9x) | 256 | 35.4 |

Table 2: **Performance comparison of various acceleration methods on Dream-Base-7B.** Speedup ratios relative to the baseline are shown in (green). The max generation length is set to 256. See Appendix D for detailed hyperparameters.

Additional benchmarking against leading auto-regressive (AR) LLMs of comparable scale, specifically LLaMA3-Instruct-8B (Grattafiori et al., 2024) and Qwen2.5-Base-7B (Yang et al., 2024a), demonstrates the efficacy of D2F in achieving faster-than-AR inference speeds.

**Quantitative Results.** As shown in Figure 1, D2F represents the first open-source dLLMs to surpass state-of-the-art AR LLMs in inference speed. The maximum generation length is set to 512 for all methods to ensure fairness. Concretely, D2F-Dream-Base-7B achieves a throughput of 119.9 tokens/s on GSM8K. This constitutes a 2.5× speedup over LLaMA3-Instruct-8B (48.0 tokens/s) and a 2.3× speedup over Qwen2.5-Base-7B (52.7 tokens/s).

D2F also significantly accelerates baseline dLLMs while maintaining equivalent average performance. As shown in Table 1, D2F-LLaDA-Instruct-8B achieves a 52.9× speedup (47.6 tokens/s vs. baseline 0.9 tokens/s) with minimal performance difference (38.0 vs. baseline 39.0), and consistently outperforms prior dLLM acceleration techniques. This owes to D2F's early termination via <EOS> detection on Instruct models, avoiding unnecessary generation. On MATH, it reaches 90.2 tokens/s, a 2.1× speedup over Fast-dLLM's Dual-Cache (42.5 tokens/s). For fairness, the baseline hyperparameters follow the settings in (Nie et al., 2025). Similarly, as shown in Table 2, D2F-Dream-Base-7B attains 91.2 tokens/s on GSM8K-CoT, corresponding to a 9.6× speedup over Dream-Base-7B (9.5 tokens/s) with slight performance improvement, and a 1.8× speedup over Fast-dLLM (49.8 tokens/s). Since Dream-Base struggles to generate the stop token correctly, we set a unified maximum length of 256 for all methods in this comparison, following the setting in dLLM-cache (Liu et al., 2025). We vary max lengths (512 vs. 256) to strictly align with respective baseline protocols for fair comparison. These results demonstrate that D2F not only surpasses existing acceleration methods but also enables dLLMs to exceed AR LLMs in throughput, substantially enhancing their practical applicability. More detailed hyperparameter settings are provided in Appendix D.

## 5.3 ABLATIONS AND ANALYSIS

This section presents ablation studies to dissect the contributions of key components within D2F. All ablation experiments are conducted on the D2F-Dream-Base-7B model unless otherwise specified.

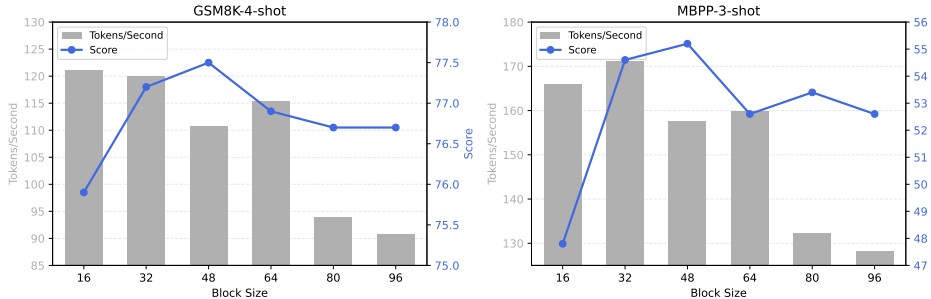

Figure 5: **Ablation study on the block size during inference.** The block size for inference is tested with integer multiples of the training block size (16). All experiments were conducted with a maximum length of 512, $\tau_{\text{conf}} = 0.9$, $\tau_{\text{add}} = 0.1$, and $\tau_{\text{act}} = 0.95$.

| $\tau_{\text{act}} = \tau_{\text{conf}}$ | $\tau_{\text{add}}$ | TPS ↑ | Score ↑ |
|---|---|---|---|
| | 0.95 | 105.2 | 76.9 |
| | 0.7 | 107.2 | 77.2 |
| 0.95 | 0.5 | 106.3 | 77.3 |
| | 0.1 | 104.0 | **77.7** |
| | 0.90 | 124.5 | 74.7 |
| | 0.7 | 126.2 | 75.7 |
| 0.90 | 0.5 | 124.7 | 76.2 |
| | 0.1 | 122.1 | **76.4** |
| | 0.85 | 136.8 | 72.6 |
| | 0.7 | 139.0 | 74.2 |
| 0.85 | 0.5 | 138.5 | 74.0 |
| | 0.1 | 135.4 | **75.0** |

Table 3: **Ablation of inference hyperparameters on GSM8K-4-shot.**

| Benchmark | $\tau_{\text{act}}$ | Model | TPS ↑ | Score ↑ |
|---|---|---|---|---|
| | 0.95 | random | 147.2 | 49.6 |
| | | D2F | **171.2** | **54.6** |
| MBPP | 0.90 | random | 148.5 | 48.6 |
| 3-shot | | D2F | **175.5** | **53.2** |
| | 0.85 | random | 150.6 | 47.6 |
| | | D2F | **177.5** | **52.6** |
| | 0.95 | random | 114.5 | 77.1 |
| | | D2F | **119.9** | **77.2** |
| GSM8k | 0.90 | random | 116.7 | **76.5** |
| 4-shot | | D2F | **123.5** | 76.4 |
| | 0.85 | random | 118.8 | **75.9** |
| | | D2F | **124.8** | 75.4 |

Table 4: **Ablation of the noise scheduling on MBPP-3-shot and GSM8k-4-shot.** Comparison of D2F and random noise schedules.

**Throughput-performance trade-off.** Figure 2 illustrates the throughput-performance trade-off for different methods on GSM8K and MBPP. D2F-Dream-Base-7B employs a fixed $\tau_{\text{add}} = 0.1$ and a block size of 32, with unified thresholds $\tau_{\text{act}} = \tau_{\text{conf}}$ being varied. Results demonstrate that D2F-Dream-Base-7B achieves significantly higher efficiency than baselines. For instance, on GSM8K, D2F-Dream-Base-7B attains 150.9 tokens/sec with a score of 71.2, achieving $3.1\times$ the throughput of LLaMA3-Instruct-8B (48.0 tokens/sec) while exceeding its score (70.1). In contrast, Dream-Base-7B exhibits substantial performance degradation at higher throughput: reducing sampling steps from 512 to 128 causes its GSM8K score to drop from 71.4 to 42.8. This demonstrates the superior capability of D2F-Dream-Base-7B in maintaining performance during accelerated inference.

**Effect of block size for inference.** Figure 5 demonstrates the influence of block size on inference performance and speed. Overall, increasing block size consistently reduces throughput while yielding an initial performance improvement followed by deterioration. Optimal block size selection moderately reduces throughput but maximizes performance—for example, a block size of 48 achieves the peak GSM8K score of 77.5, surpassing smaller block sizes such as 16.

**Ablation on inference hyperparameters.** Table 3 studies $\tau_{\text{conf}}$, $\tau_{\text{act}}$, and $\tau_{\text{add}}$. When $\tau_{\text{add}} = \tau_{\text{act}}$, new blocks are immediately activated (single-state). Our dual-state design ($\tau_{\text{add}} < \tau_{\text{act}}$) introduces a conservative initial state and consistently performs better. For example, with $\tau_{\text{act}} = 0.85$, lowering $\tau_{\text{add}}$ from 0.85 to 0.7 increases the score from 72.6 to 74.2 and throughput from 136.8 to 139.0 TPS; further reduction yields additional score gains with only slight throughput loss.

## 6    CONCLUSION

In this work, we introduce Discrete Diffusion Forcing (D2F), a novel training paradigm for dLLMs. D2F employs a generation scheme that conditions on partially predicted tokens from previous blocks to predict the next block, thereby supporting KV cache and enabling parallel generation across multiple blocks, resulting in significantly faster inference. Empirically, extensive experiments demonstrate that D2F achieves the milestone of being the first dLLM to support faster-than-AR inference.

## ETHICS STATEMENT

This work does not involve human subjects, animal experiments, or sensitive data. Therefore, it does not raise any ethical concerns.

## REPRODUCIBILITY STATEMENT

To promote reproducibility, we plan to release the source code for both training and inference of the proposed method in the future. The released code and accompanying instructions will enable other researchers to reproduce our results and build upon this work.

## ACKNOWLEDGEMENTS

This work was supported by Shanghai Key Technology R&D Program "New Generation of Information Technology" (No. 25511103700), NSF of China (Nos. 62306176, 92470118), Natural Science Foundation of Shanghai (No. 23ZR1428700), CCF-ALIMAMA TECH Kangaroo Fund (NO. CCF-ALIMAMA OF 2025010), and Ant Group.

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

## A   THE USE OF LARGE LANGUAGE MODELS

In this work, large language models (LLMs) were employed solely as writing assistants. Their role was limited to polishing the language, improving clarity, and refining the overall readability of the manuscript. No part of the conceptual development, experimental design, data analysis, or interpretation of results relied on LLMs.

## B   ABLATION STUDIES ON D2F COMPONENTS

To isolate the performance gains from different components of our D2F framework, we conduct ablation studies on both the **D2F-LLaDA** (Table 5) and **D2F-Dream** (Table 7) models. We compare two configurations:

- **Cache-only**: This configuration utilizes the block-wise causal attention mechanism to enable standard KV cache but does not employ the parallel, asynchronous block generation pipeline. Generation proceeds serially, one block at a time.
- **Cache + Para**: This is the full D2F method, combining KV cache with our pipelined parallel decoding algorithm.

The results clearly demonstrate that while enabling KV cache (Cache-only) provides a substantial speedup, the addition of our parallel decoding pipeline (Cache + Para) further accelerates inference by a significant margin (e.g., from 2.4x to 7.3x on GSM8K for LLaDA), highlighting the efficacy of the asynchronous generation strategy.

**Advantages of D2F training over fully random masks.** As shown in Table 4, the D2F training strategy significantly outperforms a baseline utilizing random noise schedules per block. On MBPP, D2F demonstrates comprehensive superiority, achieving a 5.0-point score improvement (54.6 versus 49.6) and 24.0 TPS gain at $\tau_{\text{act}} = 0.95$. These results substantiate that the structured progressive noising of D2F delivers a more effective training objective than fully random masking.

## C   CONTROL EXPERIMENT FOR DATA CONTRIBUTION

To prove that our performance gains stem from the D2F framework and not the training data, we conducted a control experiment. We created a control model, Dream-Base*, by directly fine-tuning the original Dream-Base on the same distillation dataset. For a fair comparison, this fine-tuning used LoRA with the exact same parameters as our D2F training.

The results are in Table 6. While Dream-Base* shows minor score improvements on some tasks, its inference speed drops significantly. This speed degradation is due to the computational overhead of the added LoRA layers. This control experiment confirms that our D2F methodology, rather than the data, is the key driver of the exceptional inference acceleration.

## D   HYPERPARAMETER DETAILS

Table 8 details the specific hyperparameter configurations used for both the baseline models and our D2F models across all evaluated benchmarks. Baseline settings are adopted from their respective original works to ensure fair comparison. For our D2F models, we specify the maximum generation length, the inference block size, and the key thresholds for our pipelined parallel decoding algorithm: the token addition threshold ($\tau_{add}$), the block activation threshold ($\tau_{act}$), and the token confirmation confidence ($\tau_{conf}$).

## E   ADDITIONAL EXPERIMENTAL RESULTS

We further train a D2F model based on DiffuCoder-Instruct (Gong et al., 2025). Specifically, we selected 30K samples from the SFT-stage training data of OpenCoder (Huang et al., 2024) and follow the same training configuration as D2F-Dream, training for 28,000 iterations. The resulting model, D2F-DiffuCoder, is evaluated on HumanEval, HumanEval+, and MBPP+ (Liu et al.,

| Test Set | Configuration | TPS ↑ | Latency (s) ↓ | Gen. Length | Score ↑ |
|---|---|---|---|---|---|
| **GSM8K** | Cache-only | 17.5 (2.4x) | 8.3 (3.9x) | 145 | **78.1** |
| 4-shot | Cache + Para | **52.5** (7.3x) | **2.8** (11.5x) | 144 | 77.3 |
| **MBPP** | Cache-only | 18.1 (20.1x) | 3.8 (18.8x) | 69 | 37.6 |
| 3-shot | Cache + Para | **47.6** (52.9x) | **1.4** (51.0x) | 68 | **38.0** |
| **HumanEval** | Cache-only | 28.0 (10.0x) | 4.8 (8.1x) | 135 | **40.2** |
| 0-shot | Cache + Para | **81.6** (29.1x) | **1.6** (24.3x) | 133 | **40.2** |
| **Math** | Cache-only | 30.6 (1.5x) | 12.6 (0.9x) | 385 | **29.6** |
| 4-shot | Cache + Para | **90.2** (4.3x) | **4.3** (2.7x) | 384 | 29.1 |

Table 5: Ablation study of our proposed method (**D2F-LLaDA**) on the **LLaDA-Instruct** model. Performance ratios relative to the LLaDA-Instruct baseline (from Table 1) are shown in (bright green).

| Test Set | Method | TPS ↑ | Latency (s) ↓ | Gen. Length | Score ↑ |
|---|---|---|---|---|---|
| **GSM8K-CoT** | Dream-Base | **9.5** (1.0x) | **26.8** (1.0x) | 255 | 75.0 |
| 8-shot | Dream-Base* | 6.7 (0.7x) | 29.9 (0.9x) | 199 | **77.8** |
| **MBPP** | Dream-Base | **10.4** (1.0x) | **24.6** (1.0x) | 256 | 56.2 |
| 3-shot | Dream-Base* | 4.2 (0.4x) | 27.4 (0.9x) | 114 | **57.4** |
| **HumanEval** | Dream-Base | **20.2** (1.0x) | **12.6** (1.0x) | 255 | **54.3** |
| 0-shot | Dream-Base* | 8.8 (0.4x) | 14.2 (0.9x) | 125 | 51.8 |
| **Math** | Dream-Base | **9.9** (1.0x) | **25.8** (1.0x) | 256 | **35.8** |
| 4-shot | Dream-Base* | 5.4 (0.5x) | 28.6 (0.9x) | 154 | 33.4 |

Table 6: Comparison between the baseline Dream-Base model and its fine-tuned version, Dream-Base*. The fine-tuned version, highlighted with a gray background, is obtained by directly fine-tuning on the distillation data. Performance ratios relative to the baseline are shown in (bright green), and the highest score in each test set is marked in **bold**.

| Test Set | Configuration | TPS ↑ | Latency (s) ↓ | Gen. Length | Score ↑ |
|---|---|---|---|---|---|
| **GSM8K-CoT** | Cache-only | 40.7 (4.3x) | 6.3 (4.3x) | 256 | **77.8** |
| 8-shot | Cache + Para | **91.2** (9.6x) | **2.8** (9.6x) | 256 | 77.6 |
| **MBPP** | Cache-only | 40.5 (3.9x) | 5.9 (4.2x) | 240 | 54.2 |
| 3-shot | Cache + Para | **105** (10.1x) | **2.3** (10.7x) | 240 | **55.2** |
| **HumanEval** | Cache-only | 43.6 (2.2x) | 5.1 (2.5x) | 222 | 53.7 |
| 0-shot | Cache + Para | **73.2** (3.6x) | **3.1** (4.1x) | 227 | **54.3** |
| **Math** | Cache-only | 39.6 (4.0x) | 6.5 (4.0x) | 256 | **35.9** |
| 4-shot | Cache + Para | **98.8** (10.0x) | **2.6** (9.9x) | 256 | 35.4 |

Table 7: Ablation study of our proposed method (**D2F-Dream**) on the **Dream-Base** model. Performance ratios relative to the Dream-Base baseline (from Table 2) are shown in (bright green).

| Benchmark | Configuration | length | block_size | $\tau_{add}$ | $\tau_{act}$ | $\tau_{conf}$ |
|---|---|---|---|---|---|---|
| **Method: D2F-LLaDA** | | | | | | |
| GSM8K | Baseline | 256 | 8 | – | – | – |
| | D2F | 512 | 64 | 0.7 | 0.95 | 0.9 |
| MBPP | Baseline | 512 | 32 | – | – | – |
| | D2F | 512 | 32 | 0.9 | 0.95 | 0.9 |
| HumanEval | Baseline | 512 | 32 | – | – | – |
| | D2F | 512 | 32 | 0.1 | 0.95 | 0.9 |
| MATH | Baseline | 256 | 256 | – | – | – |
| | D2F | 512 | 32 | 0.1 | 0.95 | 0.9 |
| **Method: D2F-Dream** | | | | | | |
| GSM8K-COT | Baseline | 256 | – | – | – | – |
| | D2F | 256 | 32 | 0.1 | 0.95 | 0.9 |
| MBPP | Baseline | 256 | – | – | – | – |
| | D2F | 256 | 48 | 0.1 | 0.95 | 0.9 |
| HumanEval | Baseline | 256 | – | – | – | – |
| | D2F | 256 | 32 | 0.9 | 0.95 | 0.95 |
| MATH | Baseline | 256 | – | – | – | – |
| | D2F | 256 | 64 | 0.1 | 0.95 | 0.9 |

Table 8: Sampling hyperparameters for the experiments in Table 1 and Table 2. Baseline settings are adopted from prior work. For D2F, the `length` parameter only sets a maximum generation limit without affecting the sampling distribution due to its block-wise nature. Notably, for the Dream-based model, we match D2F's maximum length to the baseline's, as the base model often fails to generate proper termination tokens.

2023), achieving up to a 10× speedup while maintaining performance comparable to the original DiffuCoder-Instruct. The results are shown in Table 9.

| Test Set | Method | TPS ↑ | Score ↑ |
|---|---|---|---|
| **HumanEval+** | DiffuCoder-Instruct | 11.05 (1.0x) | 65.2 |
| 0-shot | D2F-DiffuCoder | **82.06** (7.4x) | **65.9** |
| **MBPP+** | DiffuCoder-Instruct | 3.94 (1.0x) | **61.9** |
| 0-shot | D2F-DiffuCoder | **46.53** (11.8x) | 61.4 |
| **HumanEval** | DiffuCoder-Instruct | 11.05 (1.0x) | **72.0** |
| 0-shot | D2F-DiffuCoder | **82.06** (7.4x) | 71.3 |

Table 9: Comparison between the baseline DiffuCoder-Instruct model and D2F-DiffuCoder.(Since the HumanEval+ benchmark includes HumanEval, for convenience we report the average speed on HumanEval+ for both the HumanEval and HumanEval+ tasks.)

## F  PERFORMANCE VARIANCE AND ERROR BARS

Unlike autoregressive models or standard diffusion sampling which involve stochasticity, our D2F inference uses a greedy, confidence-based sampling algorithm (with temperature set to 0) that is deterministic. Consequently, D2F results do not exhibit variance across runs.

However, to address concerns regarding the statistical significance of our baselines, we quantify the variance of the stochastic baselines—Vanilla dLLM (Dream-Base) and dLLM-Cache (Liu et al., 2025). We conducted 3 runs on GSM8K (4-shot, 256 length) using different random seeds. As

shown in Table 10, the performance variation is minimal (standard deviation $\approx 0.3$), confirming that the improvements achieved by D2F are significant and not due to random fluctuations.

| Seed | Vanilla dLLM (Score) | dLLM-Cache (Score) |
|------|----------------------|--------------------|
| 1 | 72.5 | 72.3 |
| 2 | 72.6 | 72.7 |
| 3 | 72.0 | 72.1 |
| **Mean $\pm$ Std** | **72.37 $\pm$ 0.32** | **72.37 $\pm$ 0.31** |

Table 10: Performance variance of stochastic baselines on GSM8K (4-shot). The minimal standard deviation confirms the stability of the baseline measurements.

## G  COMPARISON WITH BLOCK DIFFUSION (SDAR)

To further justify the necessity and advantages of our D2F framework, we compare it against **SDAR** (Cheng et al., 2025), a state-of-the-art block diffusion model trained from scratch. We evaluate SDAR-4B-block32 on GSM8K (4-shot, 256 length) using the same block size as D2F.

As shown in Table 11, while SDAR achieves competitive scores, its inference speed is significantly limited. Even with reduced confidence thresholds, SDAR maxes out at 46.7 TPS. In contrast, D2F (as shown in Figure 2 and Table 7) achieves significantly higher throughput (up to 90+ TPS) with comparable accuracy.

| Model | Confidence | TPS | Score |
|-------|-----------|-----|-------|
| | 0.9 | 29.7 | 79.2 |
| SDAR-4B-block32 | 0.8 | 35.9 | 78.6 |
| | 0.7 | 39.6 | 76.8 |
| | 0.6 | 46.7 | 72.3 |

Table 11: Performance trade-off of SDAR-4B on GSM8K. Despite being a specialized block diffusion model, SDAR lacks the inter-block parallelism of D2F, resulting in lower throughput.

**Advantages of D2F over Block Diffusion models like SDAR:**

1. **Higher Inference Speed via Parallelism:** D2F's pipelined parallel decoding removes the sequential bottleneck between blocks. In SDAR, block $t+1$ must wait for block $t$ to finish; in D2F, future blocks start decoding early, leading to significantly higher TPS.

2. **Training Efficiency via Distillation:** SDAR requires expensive pre-training from scratch on massive datasets (trillions of tokens). In stark contrast, D2F adapts existing pre-trained dLLMs (e.g., Dream-7B) using only $\sim$12 hours of fine-tuning on a small dataset (17k samples). This makes D2F a far more accessible and cost-effective solution for refurbishing existing models into efficient block-wise generators.

