# OpenReview forum: "Diffusion LLMs Can Do Faster-Than-AR Inference via Discrete Diffusion Forcing"
_ICLR.cc/2026/Conference — ICLR 2026 Poster_

### Official Review · Reviewer_gxgg · 2025-10-30

**Soundness:** 3
**Presentation:** 3
**Contribution:** 2
**Rating:** 6
**Confidence:** 3

**Summary:**

This paper proposes Discrete Diffusion Forcing (D2F), a framework for faster text generation with diffusion-based large language models. D2F combines asymmetric distillation from a bidirectional teacher with block-wise causal decoding to enable partial KV caching. The authors report substantial inference speedups over both prior diffusion LLMs and standard autoregressive models while maintaining competitive accuracy on reasoning and coding benchmarks.

**Strengths:**

Clear motivation: Tackles the critical limitation of diffusion LLMs, slow inference, and provides a coherent framework aimed at bridging the efficiency gap with autoregressive models.

Technical integration: The combination of block-wise causal decoding, diffusion distillation, and pipelined inference is well-engineered and practically significant.

Strong empirical results: Shows large speedups with minimal accuracy loss.

Accessible open-source implementation: The authors promise to open source the code.

**Weaknesses:**

Limited novelty. Most structural components (block-wise causal diffusion, caching, block parallelism) are drawn from prior work. I understand the novelty comes from the asymmetric distillation and the generation of later blocks without fully generated earlier blocks?

Insufficient Justification for Asymmetric Distillation. The paper justifies distillation primarily as a way to reduce training costs. However, it doesn't provide an ablation comparing its asymmetric method to a "from-scratch" D2F model (even on a smaller scale) or a simpler "standard" distillation. This makes it hard to isolate how relevant this distillation mechanism is.

**Questions:**

KV caching explanation. One source of speedups is the use of block-wise KV caching. Without the strictly causal generation over blocks some blocks are getting unmasked with previous blocks now fully unmasked. How is the KV caching implemented here? Or is KV caching only used for fully unmasked blocks? This is related to the ablations in the Appendix (table 5 and 7) which show that the pipelined decoding leads to 2-3x speedups.

The paper introduces a form of asymmetric self-distillation to train the model. This is one of the methodological contributions of the paper. It is unclear to me how much this is needed. Does this have any benefits over training the model from scratch (apart from training cost)? Is the asymmetric component needed? You could imagine training with the teacher using exactly the same setup as the student (standard distillation); that could perform similarly? (I understand the teacher is fully bidirectional, but should be able to handle such task as well.)

---

> ### Author Response · Authors · 2025-11-20
>
> Thank you for your thoughtful feedback and constructive questions. We have clarified our contributions and addressed your concerns below:
>
> ---
>
> **W1: Clarification of the Contributions**
>
> **A:** We thank the reviewer for this insightful comment. We acknowledge that our work draws inspiration from prior ideas, including block parallelism from the video domain, and caching and block-wise causality from existing DLLMs. Here, we would like to clarify our novel contributions:
>
> - First, our work is driven by a novel goal: unlocking inter-block parallelism for acceleration. This fundamentally differs from prior Diffusion Forcing[4] work, which focused on mitigating prediction bias.
> - Second, achieving this goal required us to be the first to adapt the Diffusion Forcing principle from continuous domains to discrete text models.
> - Third, to implement this adaptation efficiently, we designed a novel asymmetric distillation framework that repurposes existing dLLMs at a low cost.
> - Finally, we co-designed a specialized pipelined sampling algorithm to fully leverage this new capability during inference.
>
> We will clarify these contributions in the revision.
>
> **Q1: Clarification of the KVcache**
>
> **A:** We apologize for the unclear description of this mechanism in the paper. To clarify, KV caching is applied only to blocks that are fully unmasked. We will explain this implementation in more detail in the revision.
>
> **W2 & Q2: The Necessity of Asymmetric Distillation**
>
> **A:** We thank the reviewer for this question. The necessity of asymmetric distillation is to create a block-wise diffusion model capable of high-throughput inference via exact KV caching and inter-block parallelism, at a cost far lower than training a model from scratch using the Block Diffusion[3].
>
> 1.  **Why Asymmetric Distillation is Required:** The original bidirectional teacher cannot use exact KV caching because it causes significant performance degradation, as noted in Fast-dllm[1] and dllm-cache[2]. Standard distillation is insufficient as it cannot alter the student's attention mechanism to be causal, making the required block-wise causal inference impossible. Our method is designed to perform this required architectural change.
>
> 2.  **Why Logits Distillation is Used:** We found that directly fine-tuning on real data to change the model's attention mechanism caused a significant performance drop. The model failed to learn the block-wise causal paradigm from limited samples. In contrast, distilling from the teacher's logits provides a much richer training signal, which is crucial for preserving the model's original performance while accelerating training.
>
> **Reference**
>
> [1] Wu, Chengyue, et al. "Fast-dllm: Training-free acceleration of diffusion llm by enabling kv cache and parallel decoding." arXiv preprint arXiv:2505.22618 (2025).
>
> [2] Liu, Zhiyuan, et al. "dllm-cache: Accelerating diffusion large language models with adaptive caching." arXiv preprint arXiv:2506.06295 (2025).
>
> [3] Arriola, Marianne, et al. "Block diffusion: Interpolating between autoregressive and diffusion language models." arXiv preprint arXiv:2503.09573 (2025).
>
> [4] Chen, Boyuan, et al. "Diffusion forcing: Next-token prediction meets full-sequence diffusion." Advances in Neural Information Processing Systems 37 (2024): 24081-24125.

---

> > ### Author Response · Authors · 2025-11-26
> >
> > We truly appreciate your helpful suggestions and have carefully answered each of your questions. If any points require additional explanation, we would be pleased to clarify. We respectfully hope that our responses justify consideration for a better score.

---

### Official Review · Reviewer_WETF · 2025-11-01

**Soundness:** 3
**Presentation:** 3
**Contribution:** 2
**Rating:** 6
**Confidence:** 4

**Summary:**

This paper presents "Discrete Diffusion Forcing" (D2F), a framework to make diffusion LLMs (dLLMs) faster than autoregressive (AR) LLMs. D2F reformulates the dLLM as an AR-diffusion hybrid, which solves the core problem of dLLMs not being able to use a standard KV cache. The method uses asymmetric distillation to train a block-wise causal student dLLM to mimic a bidirectional teacher, using a monotonically increasing noise schedule. This enables a "pipelined parallel decoding" algorithm during inference, where future blocks can be processed before previous blocks are fully finished, all while using an exact KV cache. The authors report significant speedups (up to 2.5x vs. LLaMA3, 50x vs. vanilla dLLMs) with comparable performance.

**Strengths:**

1. Significant Milestone: The paper achieves "faster-than-AR" inference with an open-source dLLM, a significant milestone. The reported speedups (2.5x vs. LLaMA3, 50x vs. LLaDA) are extremely impressive.

2. Effective Training Strategy: The "asymmetric distillation" with a "monotonically increasing mask schedule" is a clever adaptation of Diffusion Forcing to the discrete domain. It effectively trains the model to predict from an incomplete prefix, enabling the parallel pipeline.

3. Solves the KV Cache Problem: The method reframes the dLLM as a block-wise causal model, allowing the use of a standard, exact KV cache. This is a fundamental improvement over prior work that relied on approximate caching.

4. Strong Ablation Studies: The ablations clearly validate the design, showing that the parallel pipeline adds significant speed over caching alone, the structured noise schedule is critical, and the gains are not just an artifact of the data or tuning.

**Weaknesses:**

1. Inconsistent/Confusing Performance Claims: The paper's headline claim of "faster-than-AR" performance is made confusing by seemingly inconsistent numbers across the text and figures. This makes the exact performance trade-off difficult to assess.

- LLaMA3 Baseline: In Figure 2, the LLaMA3-Instruct-8B baseline (star) is plotted with a GSM8K score of ~77. However, the text in Section 5.3 states its score is 70.1. This is a significant discrepancy.

- D2F Performance: The paper reports multiple different performance points for its own model. Figure 1 claims 119.9 TPS (which corresponds to a score of ~75 in Figure 2). But Section 5.3 reports a different operating point of 150.9 TPS and a 71.2 score.

- This inconsistency makes it difficult to definitively conclude if D2F is "faster and comparable," "faster and slightly worse," or "much faster and slightly better" than its key AR competitor.

2. Inference Complexity: The proposed "pipelined parallel decoding" (Algorithm 2) introduces significant new complexity to the inference process. It requires careful tuning of three new, interacting hyperparameters ($\tau_{add}$, $\tau_{act}$, $\tau_{conf}$) in addition to the block size, which could be a barrier to practical adoption and tuning compared to the simplicity of standard AR decoding.

3. Incremental Novelty: The paper is transparent that its core idea is an "extension of DF (Diffusion Forcing)" (Chen et al., 2024a) and "connects to CausVid" (Yin et al., 2025). While the adaptation to the discrete domain is novel and the engineering is strong, the fundamental concept of forcing a model to predict from a noisy prefix via distillation is not entirely new. This makes the contribution more of a highly successful and non-trivial adaptation than a completely new paradigm.

**Questions:**

1. Could the authors please clarify the performance of the LLaMA3-Instruct-8B baseline on GSM8K? Figure 2 implies a score of ~77, but the text in Section 5.3 states 70.1. Which number should be used for comparison, and why is there a discrepancy?

2. Similarly, could the authors provide a single, clear {TPS, Score} pair for the D2F-Dream-Base-7B model on GSM8K that represents the main claim? The numbers in Figure 1, Figure 2, and Section 5.3 all seem to refer to different operating points, making a direct comparison difficult.

3. The inference pipeline (Alg. 2) seems complex. How sensitive is the model's performance (both speed and quality) to the choice of $\tau_{add}$, $\tau_{act}$, and $\tau_{conf}$? Table 3 suggests they interact, but is there a "default" set of parameters that works well across most tasks?

---

> ### Author Response · Authors · 2025-11-20
>
> Thank you for your thoughtful feedback and constructive questions. We have addressed your concerns below:
>
> ---
>
> **W1 & Q1: Clarification on LLaMA3's Performance in Figure 2**
>
> **A:** We thank the reviewer and would like to clarify the presentation in Figure 2. The performance of LLaMA3 on GSM8K is 70.1. In Figure 2, this result is represented by the purple star. This star is positioned approximately midway between the 65 and 75 y-axis tick marks, which is consistent with the 70.1 score reported in Section 5.3.
>
> **Q2: Clarification of the result of D2F**
>
> **A:** We apologize for the confusion. The different D2F results stem from distinct experimental setups, each chosen to ensure a specific, fair comparison.
>
> Our main results in Table 2 (GSM8K-COT 8-shot, 256 length) use a unified protocol for a fair comparison against prior baselines like dLLM-Cache[1], which used the same setting.
>
> Conversely, the comparisons in Figure 1 and Section 5.3 use a different setting (GSM8K 4-shot, 512 length) for a unified and fair comparison against a broader set of models. The multiple points for D2F in Figure 2 specifically map out its speed-performance trade-off by varying decoding thresholds, as discussed in Section 5.3, which explains why they differ from the single result in Figure 1.
>
> We will clarify the motivation for each setting in the revision.
>
> **W2 & Q3: Selection of Inference Hyperparameters**
>
> **A:** Thank you for this question. A strong default configuration of {`τ_add`=0.1, `τ_act`=0.95, `τ_conf`=0.9} performs well across most tasks, making complex tuning unnecessary.
>
> As shown in Table 8, the two most impactful hyperparameters, `τ_act` and `τ_conf`, are highly stable, with values of 0.95 and 0.9 respectively being used for most experiments. Furthermore, Table 3 demonstrates that `τ_add` has only a minor impact on results, with 0.1 being a consistently effective value.
>
> We acknowledge that minor tuning can yield marginal gains on specific tasks, which explains the slight variations in our tables. We are currently exploring more adaptive methods to automatically control these parameters in future work.
>
> **W3: Clarification of the contribution**
>
> **A:** We acknowledge that our work draws inspiration from Diffusion Forcing[2] and CausVid[3]. However, our fundamental purpose is to enable inter-block parallel generation. In pursuing this, we found the model must learn to predict from noisy prefixes, a capability similar to Diffusion Forcing. While this led us to extend the DF method to discrete models, our motivation of achieving parallelism is fundamentally different from DF's original goal of mitigating prediction bias.
>
> Based on this distinct objective, we designed a asymmetric distillation framework with specialized masking strategies to prevent performance degradation during training. We also developed a corresponding sampling algorithm specifically to exploit the inter-block parallel capabilities. These tailored training and inference components are what distinguish D2F from previous work in the video generation field.
>
> We will clarify the contributions more explicitly in the revision.
>
> **Reference**
>
> [1] Liu, Zhiyuan, et al. "dllm-cache: Accelerating diffusion large language models with adaptive caching." arXiv preprint arXiv:2506.06295 (2025).
>
> [2] Chen, Boyuan, et al. "Diffusion forcing: Next-token prediction meets full-sequence diffusion." Advances in Neural Information Processing Systems 37 (2024): 24081-24125.
>
> [3] Yin, Tianwei, et al. "From slow bidirectional to fast causal video generators." arXiv e-prints (2024): arXiv-2412.

---

> > ### Author Response · Authors · 2025-11-26
> >
> > We are very grateful for your constructive comments and have provided responses to your questions. If there are any additional concerns that require clarification, please let us know. We hope that our responses provide sufficient reasons to raise the score.

---

### Official Review · Reviewer_QtKg · 2025-11-01

**Soundness:** 3
**Presentation:** 3
**Contribution:** 2
**Rating:** 6
**Confidence:** 3

**Summary:**

This paper proposes discrete diffusion forcing that critically enables decoding of the new block to start even before the previous blocks are fully decoded. This unlocks the parallelization not only within a single block but also across blocks. By further inheriting KV-cache brought by block decoding, D2F achieves promising speedup without sacrificing much sample quality. Experiments on math and coding benchmarks over two models demonstrate the effectiveness of the proposed approach in terms of achieving more speedup while maintaining the accuracy.

**Strengths:**

1. The paper tackles a critical problem in discrete diffusion model: inference acceleration. D2F unlocks parallelization even across different blocks, which is a very important feature to significantly enhance the speed.

2. The method includes a distillation training for D2F and a customized inference procedure. The distillation helps mitigate the bias in block diffusion that requires the previous blocks to be fully decoded.

3. The presentation is clear and the proposed approach is clean and very easy to follow.

**Weaknesses:**

1. The technical novelty is somewhat bounded since the work is an adaptation of previous diffusion forcing literature (especially video diffusion) to discrete diffusion models. The concern is not significant though, given the promising empirical performance.

2. Several experiment results that are key to compare the accuracy-efficiency frontier and understand the design of the proposed approach are missing. See Q1 and Q2 for more details.

**Questions:**

1. It would be more helpful to elucidate the speedup by showing the accuracy-speedup curves for the proposed approach and the baselines, instead of only showing one single point on the curve (the numbers in the table). i.e, plot the curve of Fast-dLLM and dLLM-Cache, if applicable.

2. How is the accuracy-efficiency tradeoff of Block Diffusion [1]? This can be investigated by doing block diffusion distillation and sweep over different numbers of decoding steps at inference time. This is important for ablating the necessary of diffusion forcing distillation over naive block diffusion distillation.


[1] Arriola et al. Block diffusion: Interpolating between autoregressive and diffusion language models.

---

> ### Author Response · Authors · 2025-11-20
>
> Thank you for your thoughtful feedback and constructive questions. We have clarified our contributions and addressed your concerns below:
>
> ---
>
> **W1: Clarification of the Contributions**
>
> **A:** We thank the reviewer for this comment and clarify our novel contributions below:
>
> - **Adaptation to Discrete Domain:** We are the first to successfully adapt the Diffusion Forcing[5] principle from continuous domains to discrete diffusion models for text generation.
> - **Distinct Motivation:** Our motivation is fundamentally different. Whereas prior work used this approach to mitigate prediction bias, our primary goal is to unlock **inter-block parallelism** for inference acceleration.
> - **Novel Distillation Framework:** To achieve this goal, we propose a novel asymmetric distillation framework specifically tailored to train discrete diffusion models for this parallel decoding capability.
> - **Co-designed Inference Algorithm:** Finally, to fully leverage this new capability, we co-designed a specialized pipelined sampling algorithm that realizes the inter-block parallelism in practice.
>
> We will clarify these contributions more explicitly in the revision.
>
> **Q1: Accuracy-Speedup Trade-off Curves**
>
> **A:** Thank you for this important suggestion. To enable a direct comparison, we generated accuracy-speedup curves for Fast-dLLM[3] and dLLM-Cache[4] under consistent settings (sequence length=512) on GSM8K (4-shot) with Dream-Base.
>
> **Fast-dLLM (varying confidence threshold):**
>
> | Confidence | Score | TPS |
> |:----------:|:-----:|:---:|
> |    0.9     | 74.1  | 44.5|
> |    0.8     | 72.0  | 48.7|
> |    0.7     | 69.1  | 53.5|
> |    0.6     | 65.0  | 58.2|
>
> Fast-dLLM exhibits a poor trade-off: a moderate 1.3× speedup comes at the cost of a significant 9.1-point accuracy drop, as lowering the confidence threshold quickly degrades performance.
>
> **dLLM-Cache (varying step ratio):**
>
> | Steps/Length | Score | TPS |
> |:------------:|:-----:|:---:|
> |   512/512    | 68.3  | 18.6|
> |   256/512    | 58.2  | 36.1|
> |   128/512    | 38.5  | 71.6|
>
> The strategy of naively reducing inference steps is ineffective for dLLM-Cache, leading to a catastrophic performance collapse (a 29.8-point decline).
>
> In stark contrast, **D2F achieves a far superior trade-off, reaching a 70.7 score at 154.4 TPS** (Figure 2). This single data point surpasses both baselines in speed and accuracy, confirming that our inter-block parallelization is a fundamentally more effective acceleration method. We will add a comprehensive trade-off curve to the final version to visualize these advantages clearly.
>
> **Q2: Accuracy-Speedup Trade-off for Block Diffusion[1]**
>
> **A:** Thank you for this important suggestion. To justify our distillation approach, we evaluated the SOTA block-diffusion model, **SDAR-4B-block32[2]** on GSM8K (4-shot, 256 length) with the same block size as D2F for a direct comparison.
>
> | Confidence | Score | TPS |
> |:----------:|:-----:|:---:|
> |    0.9     | 79.2  | 29.7|
> |    0.8     | 78.6  | 35.9|
> |    0.7     | 76.8  | 39.6|
> |    0.6     | 72.3  | 46.7|
>
> The results offer two key insights. First, **D2F is significantly faster** because its pipelined parallelization removes the sequential bottleneck between blocks—a limitation in SDAR. Second, although SDAR achieves slightly higher accuracy, this reflects its **costly pre-training from scratch** on massive datasets. D2F, by contrast, is adapted from a 7B teacher using only **9,000 fine-tuning samples**.
>
> This comparison confirms the necessity of our distillation method: it provides an efficient way to **repurpose existing dLLMs**, bypassing the expensive pre-training required by standard block diffusion[1] models like SDAR.
>
> **Reference**
>
> [1] Arriola, Marianne, et al. "Block diffusion: Interpolating between autoregressive and diffusion language models." arXiv preprint arXiv:2503.09573 (2025).
>
> [2] Cheng, Shuang, et al. "SDAR: A Synergistic Diffusion-AutoRegression Paradigm for Scalable Sequence Generation." arXiv preprint arXiv:2510.06303 (2025).
>
> [3] Wu, Chengyue, et al. "Fast-dllm: Training-free acceleration of diffusion llm by enabling kv cache and parallel decoding." arXiv preprint arXiv:2505.22618 (2025).
>
> [4] Liu, Zhiyuan, et al. "dllm-cache: Accelerating diffusion large language models with adaptive caching." arXiv preprint arXiv:2506.06295 (2025).
>
> [5] Chen, Boyuan, et al. "Diffusion forcing: Next-token prediction meets full-sequence diffusion." Advances in Neural Information Processing Systems 37 (2024): 24081-24125.

---

> > ### Author Response · Authors · 2025-11-26
> >
> > We are deeply appreciative of your constructive guidance and have provided complete responses to your questions. If any points would benefit from further explanation, we would welcome the chance to clarify. We respectfully hope that our responses might merit your consideration for an enhanced evaluation.

---

### Official Review · Reviewer_au9N · 2025-11-01

**Soundness:** 3
**Presentation:** 3
**Contribution:** 3
**Rating:** 6
**Confidence:** 4

**Summary:**

The paper proposes Discrete Diffusion Forcing (D2F). At inference time, the framework combines the strengths of diffusion LLMs (dLLMs) and autoregressive LLMs (AR LLMs). It adopts an AR‑like blockwise sequential generation scheme to exploit the KV cache while preserving parallel decoding across blocks in dLLMs, so later tokens can be predicted without waiting for the previous block to finish. This hybrid of AR and diffusion substantially accelerates dLLM inference. During training, the method performs asymmetric distillation from a pretrained dLLM teacher with standard bidirectional attention to a student that has only a causally restricted view. Experiments on multiple benchmarks show that D2F significantly speeds up dLLM inference beyond AR LLMs while maintaining accuracy.

**Strengths:**

1) The work introduces an original hybrid paradigm that enables KV‑cache friendly, AR‑style generation while retaining dLLMs’ cross‑block parallelism, and it provides a tailored asymmetric distillation procedure to train the model.
2) According to the paper, D2F yields the first open‑source dLLMs that surpass state‑of‑the‑art AR LLMs in inference speed, and achieves more than 10× speedup on some benchmarks over dLLM baselines without D2F.
3) The experimental study is comprehensive, with comparisons against strong AR LLM and dLLM baselines on common benchmarks, along with ablations on key hyperparameters.

**Weaknesses:**

1) Training relies on a pretrained dLLM as the teacher, which may limit scaling to stronger future D2F variants. In addition, D2F does not accelerate training, so the compute cost remains substantial.
2) The main figure (Figure 3) is information‑sparse; a clearer depiction of the asymmetric distillation would improve readability. Table 3 could be half‑width, since the current layout leaves excessive white space.

**Questions:**

1) Can the main experiments include error bars to establish statistical significance?
2) Tables 1 and 2 show large variation in speedup across different benchmarks and different dLLM base models for D2F, for example 52.9× versus 4.3×. What explains this variance? Does it indicate sensitivity to model architecture or to the data type used at inference time?
3) How sensitive is D2F to generation length?

---

> ### Author Response · Authors · 2025-11-20
>
> Thank you for your thoughtful feedback and constructive questions. We have clarified our contributions and addressed your concerns below:
>
> ---
>
> **W1: Analysis of the D2F Training Method**
>
> **A:** We thank the reviewer for this important point. We acknowledge that D2F's performance is currently limited by its teacher model due to our distillation-based approach. We are actively investigating training-from-scratch alternatives for future work.
>
> Regarding the training cost, our distillation method is highly efficient. As detailed in our paper (Lines 97-98), by employing LoRA, the 7B model can be trained in only 12 hours on 8 A100 GPUs, demonstrating the low computational overhead of our approach.
>
> **W2: Explanation of the Figures and Tables**
>
> **A:** Thank you very much for your suggestions. In Figure 3, we aim to illustrate that the teacher model employs bidirectional attention while the student model uses block attention. The student model is trained by computing the KL loss between the predictions of the student and teacher models. Additionally, we will adjust the formatting of Table 3 in the revision to eliminate excessive white space.
>
> **Q1: The statistical significance of the result**
>
> **A:** We thank the reviewer for this important question.
>
> Our D2F and Fast-dLLM[1] results are based on a greedy, confidence-based sampling algorithm with temperature set to 0. This procedure is deterministic, and therefore does not produce error bars.
>
> However, we acknowledge that our baseline (vanilla dLLM) and dLLM-Cache[2] method involve stochastic sampling and thus exhibit variance. To quantify this, we conducted additional experiments on GSM8K (4-shot, 256 length) on Dream-Base model using multiple random seeds:
>
> | Seed | Baseline Score (Vanilla) | dLLM-Cache Score |
> |:----:|:------------------------:|:----------------:|
> |  1   |           72.5           |       72.3       |
> |  2   |           72.6           |       72.7       |
> |  3   |           72.0           |       72.1       |
>
> The results confirm that the performance variation for these models is minimal. For completeness, we will add error bars for the vanilla dLLM and dLLM-Cache results in the revision.
>
> **Q2: Different acceleration speed for different model and task**
>
> **A:** We thank the reviewer for this insightful question. We acknowledge that the acceleration varies across different models and tasks, and attribute this to two main factors:
>
> - **Early Termination on Instruct Models.** D2F’s block-wise generation can detect the final `<EOS>` token from Instruct models and terminate early, avoiding unnecessary generation. This yields a significant speedup over vanilla models that generate until the maximum length. This benefit is less pronounced on Base models, which typically lack an explicit `<EOS>` stopping signal.
> - **Task-Dependent Model Confidence.** On more challenging tasks, the model's prediction confidence is lower. Consequently, our confidence-based sampling accepts fewer tokens per step, which reduces the overall acceleration.
>
> We will add a detailed discussion of these factors to the revision.
>
> **Q3: Performance Across Different Generation Lengths**
>
> **A:** Thank you for the question. D2F's performance and speed are relatively insensitive to the generation length.
> This is due to two factors. First, thanks to our block-wise generation and early stopping mechanism, the maximum generation length setting has a minimal impact on D2F's performance and speed. Second, while actual output lengths vary by task, our model decodes a stable average of 2–3 tokens per inference step, which results in a relatively stable speed across different sequence lengths.
>
> **Reference**
>
> [1] Wu, Chengyue, et al. "Fast-dllm: Training-free acceleration of diffusion llm by enabling kv cache and parallel decoding." arXiv preprint arXiv:2505.22618 (2025).
>
> [2] Liu, Zhiyuan, et al. "dllm-cache: Accelerating diffusion large language models with adaptive caching." arXiv preprint arXiv:2506.06295 (2025).

---

> > ### Author Response · Authors · 2025-11-26
> >
> > We sincerely appreciate your insightful feedback and have answered the questions you raised. If there are any remaining concerns that need further clarification, we welcome the opportunity to provide additional explanations. We sincerely hope that our responses have successfully addressed your questions and can support a more positive evaluation.

---

### Author Response · Authors · 2025-11-27
**General Response**

Dear Reviewers and Area Chairs,

We sincerely thank you for your thoughtful reviews and constructive feedback throughout the review process. Your insights have been invaluable in helping us improve our work.

We have carefully addressed all the concerns and suggestions raised in the reviews, and have revised our paper accordingly. The updated manuscript incorporates your recommendations, with all changes clearly marked in red for easy identification.

We believe these revisions have significantly enhanced the quality, clarity, and contribution of our paper. If you have any remaining questions or would like further clarification on any aspect of our work, please don't hesitate to reach out. We are happy to engage in additional discussion.

Thank you once again for your time and expertise.

Best regards,

The Authors

---

### Author Response · Authors · 2025-11-29

Dear Reviewers, AC, SAC, and PC,

We sincerely appreciate the time and effort you have devoted to reviewing our manuscript. We are especially grateful for the reviewers' recognition of the novelty of our discrete diffusion forcing framework, as well as the significance of our "faster-than-AR" inference results. For clarity and convenience, we summarize below the key strengths highlighted by the reviewers, followed by our corresponding revisions and responses to their concerns.

(*We refer to Reviewer au9N as R1, Reviewer QtKg as R2, Reviewer WETF as R3, and Reviewer gxgg as R4)

**Key strengths noted by the reviewers:**
**S1: Significant Milestone**
[R1, R3, R4] The paper achieves "faster-than-AR" inference with an open-source dLLM, recognized as a significant milestone with impressive speedups (up to 50x) over vanilla dLLMs.

**S2: Effective Methodological Design**
[R2, R3] The adaptation of Diffusion Forcing to the discrete domain via asymmetric distillation is recognized as a clever strategy that effectively solves the KV cache compatibility issue.

**S3: Strong Empirical Performance**
[R1, R4] The experiments are comprehensive, showing consistent improvements across multiple benchmarks (GSM8K, HumanEval, MBPP) with strong engineering quality.

**Key responses and revisions to the comments:**
We have comprehensively responded to the reviewers' feedback and revised the manuscript accordingly. The key responses and revisions are summarized below:

* **[R2, R3, R4] (Revision):** We have explicitly clarified our contributions at the end of $\color{red}{\text{Section 1}}$. We distinguish our motivation (unlocking inter-block parallelism) from prior Diffusion Forcing work (bias mitigation) and highlight our specific adaptation to the discrete domain.
* **[R2] (Revision):** We have updated $\color{red}{\text{Figure 2}}$ to include the throughput-performance trade-off curves for Fast-dLLM and dLLM-Cache. This demonstrates that D2F achieves a more favorable trade-off frontier.
* **[R2] (Revision):** We added $\color{red}{\text{Appendix G}}$ and $\color{red}{\text{Table 11}}$ to provide a direct comparison with the Block Diffusion model (SDAR). The results demonstrate that D2F achieves significantly higher throughput (up to 90+ TPS vs 46.7 TPS) due to inter-block parallelism.
* **[R1] (Revision):** We added $\color{red}{\text{Table 4}}$ in $\color{red}{\text{Section 5.3}}$ to optimize the page layout and address the issue of excessive white space in $\color{red}{\text{Table 3}}$.
* **[R1] (Revision):** We added $\color{red}{\text{Appendix F}}$ and $\color{red}{\text{Table 10}}$ to report the performance variance and error bars for stochastic baselines (Vanilla dLLM and dLLM-Cache), confirming the statistical significance of our results.
* **[R1] (Revision):** We revised $\color{red}{\text{Section 5.2}}$ to explain the speedup variance. We explicitly explain that the high acceleration on Instruct models owes to early termination via `<EOS>` detection, whereas Base models require generating to the full length.
* **[R3] (Revision):** We clarified in $\color{red}{\text{Section 5.2}}$ and $\color{red}{\text{Tables 1 and 2}}$ that different maximum generation lengths (512 vs. 256) were used in different comparisons to strictly align with the respective baseline protocols for fairness.
* **[R4] (Response):** We addressed the concern regarding the necessity of Asymmetric Distillation. We clarified that standard distillation cannot alter the student's attention mechanism to be causal (which is required for KV cache), making our asymmetric approach essential.
* **[R3] (Response):** We clarified the confusion regarding LLaMA3's performance in $\color{red}{\text{Figure 2}}$. The position of the star in $\color{red}{\text{Figure 2}}$ aligns with the reported score of 70.1 in the text.
* **[R3] (Response):** We addressed the concern about hyperparameter sensitivity, confirming that the default configuration remains stable across most tasks.

All the revised contents are highlighted in the updated manuscript.

---

### Meta-Review · Area_Chair_vMNi · 2025-12-31

**Summary:**

This paper introduces Discrete Diffusion Forcing (D2F), an AR-diffusion hybrid for diffusion LLMs that enables KV-cache compatibility and inter-block parallel decoding via asymmetric distillation, achieving faster-than-AR inference while maintaining comparable quality. Extensive experiments show substantial speedups over both AR LLMs and prior dLLMs across reasoning and coding benchmarks.

Strengths
* Achieves a notable milestone: open-source dLLMs with faster-than-AR inference and large speedups.
* Clean, well-motivated design combining block-wise causality, KV caching, and pipelined parallel decoding.
* Strong empirical results with thorough ablations and clarified accuracy–throughput trade-offs.
* Practical training strategy (asymmetric distillation) that converges quickly and enables the required causal attention.

Recommendation
Accept. Despite concerns about novelty and reliance on distillation, the technical execution, clear practical impact, and convincing empirical gains outweigh these issues. I do agree with Reviewer gxgg that the submission would significantly benefit from additional ablation studies around training from scratch as opposed to the proposed distillation approach.

**Reviewer Concerns:**

Reviewer concerns that were mostly addressed:
* Novelty is incremental, building on prior diffusion-forcing and block-parallel ideas.
* Relies on distillation from a pretrained teacher, with added inference complexity and tuning knobs.
* Some evaluation scope and clarity concerns were raised initially, but largely addressed in rebuttal.

**Reviewer Scores:**

It's challenging to identify exactly which reviewer would change their score. All reviewers started at “weak accept / marginal accept”, not rejection. Most concerns were engineering clarity, novelty framing, and justification, not correctness. So, I anticipate that a couple of the reviewers would increase their score to 7.

---

### Decision · Program_Chairs · 2026-01-26

Accept (Poster)